# What Matters for Job Security? Exploring the Relationships among Symbolic, Instrumental Images, and Attractiveness for Corporations in South Korea

**Juyeon Oh [1] and Seunghwan Myeong [2,*]**

1   Center for Industrial Security & e-Governance, Inha University, Incheon 22212, Korea;
    juyeonoh07@gmail.com
2   Department of Public Administration, Inha University, Incheon 22212, Korea
*   Correspondence: shmyeong@inha.ac.kr

**Abstract:** This study examines the extent to which the symbolic and instrumental images and attractiveness toward an organization are related. This study further focuses on global human resource management and reports findings from two studies: Study 1 uses the data from undergraduate students, and Study 2 cross-validates the findings of Study 1 by using actual data from employees. The distinction of this study from previous works is that the present work focuses on a Korean organizational context (collectivistic cultures) and the differences between the potential applicants and employees in the perception of an organization's attractiveness. Furthermore, it investigates the relationship between the symbolic and instrumental images toward organizations, unlike existing relative research. The results show that the symbolic and instrumental images are related, and the perceptions of the corporate image differ for the potential applicants and employees in the context of collectivistic cultures. The more competent employees consider their organization to be, the more job security they perceive their organization to provide. Moreover, the symbolic image of being competent is negatively related to the instrumental image of job security. Since this study used cross-sectional data, future studies need to use longitudinal data to establish our model's causal claim empirically and investigate the underlying reasons behind these differences.

**Keywords:** recruitment; corporate image; symbolic image; instrumental image; attractiveness; South Korea

## 1. Introduction

Attracting and retaining the most competent human capital is crucial for organizational success and survival [1,2]. Since companies are aware of the fact that competent human resources are among the most valuable intangible assets [3], they make tremendous efforts to increase the effectiveness of the recruitment process, using different mediums, such as a realistic job preview (RJP), survey, and recruiting metrics [4,5]. Since hiring competent employees is an important issue, it is crucial to understand why and how potential applicants are attracted to organizations in the recruitment process. As a result, it may affect the actual application intention [6,7]. Therefore, understanding the potential applicants is essential for hiring quality employees. However, prior studies concerning recruitment have mainly concentrated on instrumental images, such as salary, location, and firm size [8,9].

People hold various images of organizations besides instrumental images. Some of these images or attributes are symbolic images [10]. Symbolic images describe the organization subjectively, but instrumental images describe the organization objectively. For instance, potential applicants consider an organization attractive because it provides a high salary (one of the instrumental images), and, at the same time, they can consider the organization to be more successful (one of the symbolic images). Hence, the organi-

zation's symbolic images can affect an individual's evaluation of the organization as an attractive workplace.

A corporate image is the result of an evaluation process [11].

Since corporate images affect purchasing decision making [12,13], retaining talent [8–10], social support in an organization [14,15], and customer loyalty [16–19], they should be managed by corporate managers. Specifically, if customers feel attracted to organizational images, they want to buy these corporates' products. For example, Mr. Park, who runs a chicken restaurant in Seoul, became known when the franchise CEO Mr. Kim Kim released a letter on his SNS. One high school student's handwritten letter contained a message expressing gratitude to Mr. Park for serving chicken to him and his younger brother, which only cost 5000 won last year. Afterward, a high school student's younger brother visited Park's chicken restaurant several times, and each time Park gave him chicken for free. When Mr. Park's story became known, consumers who wanted to give praise to Mr. Park ordered a lot because of Mr. Park's good deeds [20].

For this reason, this study concentrates on the attractive images of a corporation that can influence the application intentions of potential applicants and employees.

Several studies [2,10,21–27] supported the instrumental–symbolic framework for recruitment. However, previous works conducted studies on the relation of image factors to applicants' perceived attractiveness and concentrated mainly on western countries with individualistic cultures.

Previous research on an instrumental–symbolic framework for recruitment has been criticized for their focus on western countries, namely the United States and Belgium [2,23,28]. It is suggested that collectivist cultures are more likely to prefer symbolic work attributes (i.e., employer prestige), whereas individualistic cultures prefer instrumental attributes, such as the likelihood of advancement and remuneration [29]. However, results vary across studies. In a consumer-related study, identification does not significantly affect cultural differences [30]. A similar result is also found by Van Hoye and colleagues [2], contrary to their expectations. Van Hoye and colleagues [2] examined whether both instrumental (working conditions) and symbolic image (competence) were related to organizational attractiveness in a non-Western collectivistic culture and supported that the instrumental–symbolic framework should be applied differently across different countries and cultures. There are no consistent findings across the different cultures, namely in collectivism; therefore, we should identify the differences of the relationships among corporate images and organizational attractiveness between individualistic and collectivist cultures. Moreover, these studies were focusing on the effect of symbolic and instrumental factors on organizational perception. This study attempts to understand further what the symbolic and instrumental images of Korean companies are to comprehend the underlying reason behind the ambiguity of the symbolic–instrumental framework studies' cultural effect.

The distinction of this study from previous works is that the present work focuses on a Korean organizational context (collectivistic culture) and the differences between the potential applicants and employees in the perception of an organization's attractiveness. Furthermore, it investigates the relationship between the symbolic and instrumental images toward organizations, unlike existing relative research. This study addresses the following three research questions:

RQ1: What are the dimensions of the symbolic and instrumental images of Korean companies?

RQ2: What are the relations between symbolic images and instrumental images toward organizations?

RQ3: What are the differences in types of image factors that the potential applicants and employees consider to be attractive in South Korea?

## 2. Theoretical Background and Research Hypotheses

*2.1. Symbolic, Instrumental Images, and Attractiveness Toward the Organization*

Lievens and Highhouse [10] developed the instrumental–symbolic framework for recruitment using marketing theory [31]. The instrumental–symbolic framework concept describes the extent to which applicants consider organizations to be attractive because of instrumental and symbolic attributes.

These instrumental and symbolic attributes have more attractiveness in the field of recruitment. The instrumental images describe the objective, physical, and tangible attributes, such as salary, location, and job security. Lievens and Highhouse [10] defined instrumental images as pay, advancement, job security, task demands, location, and working with customers from the final year students' samples. Concerning data from employees, instrumental images were identified as pay, advancement, job security, task demands, benefit, and flexible working hours in Belgium. Moreover, Lievens et al. [23] used a sample of Belgian Army applicants and a sample of military employees and defined instrumental images as team/sports, structure, advancement, travel, pay, and job security. Van Hoye and colleagues [2] identified the four instrumental image dimensions in Turkey (a non-Western collectivistic culture) as pay/security (e.g., "ABC company offers above-average pay"), advancement (e.g., "ABC company offers fair opportunities for advancement"), task demands (e.g., "ABC company offers challenging tasks"), and working conditions (e.g., "ABC company offers flexible working arrangement").

On the other hand, symbolic images describe subjective, abstract, and intangible attributes, such as sincerity, competence, and excitement. Lievens et al. [23] measured symbolic images in terms of sincerity, excitement, competence, prestige, and ruggedness. Schreurs, Druart, Proost, and Witte [32] also described similar symbolic images. According to Oh and Kim [25], the five factors were sincerity, competence, affection, excitement, and sophistication. Van Hoye and colleagues [2] identified the four symbolic image dimensions to be sincerity (e.g., "honest"), innovativeness (e.g., "daring"), competence (e.g., "intelligent"), prestige (e.g., "prestigious"), and robustness (e.g., "strong").

There are various factors that may influence potential applicants' attractiveness toward organizations. Organizational characteristics, such as large-sized, decentralized, and multinational organizations [9]; the instrumental images [10,23,27]; and the symbolic images [10,23,27,32], can be considered as the factors defining the attractiveness toward an organization. Lievens and Highhouse [10] showed that the instrumental images (pay, advancement, location, working with the customer) and the symbolic images (innovativeness, competence) are related to a company's attractiveness in a student sample ($n = 261$). Moreover, the instrumental images (job security, benefits) and the symbolic images (innovativeness, competence) are related to a company's attractiveness in an employee sample ($n = 113$).

In particular, Lievens and colleagues [23] used a sample of 258 army applicants and 179 military employees. As a result, the instrumental images, such as team/sports, structure, and job security, and the symbolic images, such as excitement, competence, and ruggedness, were related to an applicant's attractiveness toward the army. In Korac, Saliterer, and Weigand's review on public sector employment [33], both instrumental images, such as job security, and symbolic images, such as trust and social responsibility, were related to employment preference.

Schreurs and colleagues [32] examined the moderating effects of applicant personality in the relationship between the symbolic images and organizational attractiveness in Belgium. The results showed that the coefficients of sincerity, prestige, and competence were significant; conscientiousness moderated the relationship between sincerity and organizational attractiveness; openness to experience moderated the relationship between excitement and organizational attractiveness.

Van Hoye and Saks [27] investigated perceptions of the organizational image and attractiveness for the Belgian Defense and the person (e. g., friend, parents) accompanying them to a job fair. The results showed that social activities and advancement were positively

related, and the structure was negatively related to potential applicants' perceived attractiveness. The symbolic images (sincerity, excitement, and prestige) were positively related, and ruggedness was negatively related to the potential applicant's perceived attractiveness. Interestingly, the educational opportunities were positively related to the perceived attractiveness of an organization. Sincerity and ruggedness were positive predictors of organizational attractiveness.

Recently, Carpentier et al. [34] found that the perceived communication characteristics of the social media of companies for the potential applicants who are looking for an actual job posting are positively related to the organizational attractiveness and word-of-mouth based on the theory of symbolic attraction. Specifically, potential applicants who visited the company's social media page can perceive the organizational warmth and competence and then feel the organizational attractiveness.

Kumari and Saini [35] examined the effect of two instrumental images (career growth opportunities, work–life balance) and one symbolic image (CSR reputation) on the employer attractiveness. As a result, they found that all factors are related to the employer's attractiveness for potential candidates.

Waples and Brachle [36] predicted that information of the organization's CSR activity would increase the organization's attractiveness and this research's data supported this hypothesis. Specifically, job seekers prefer organizations that are doing CSR activities and included this information in the recruiting materials. This study will use the instrumental–symbolic framework, which is related to organizational attractiveness, as mentioned above.

*2.2. Hypotheses*

As discussed above, many types of research on recruitment have focused on western countries, considering individualistic cultures. Van Hoye and colleagues [27] recently investigated the organizational image and attraction in a non-Western collectivistic culture. Turkish university student samples showed that the instrumental images (working conditions) and the symbolic images (competence) were positively related to a company's attractiveness.

The previous studies examined the relation of the image factors to applicants' perceived attractiveness. However, there are some limitations in a study of the relationship between symbolic and instrumental images. Moreover, prior studies have mainly concentrated on individualistic cultures. Therefore, in this study, we will investigate the dimension of Korean companies' images (collectivistic cultures) and the differences in perceptions between the potential applicants' and employees' attractiveness toward organizations. Additionally, we will investigate the relationship between the symbolic and instrumental images for organizations. Both Study 1 and Study 2 investigate the Hypotheses 1 and 2:

**Hypothesis 1 (H1).** *Symbolic images will be related to instrumental images.*

**Hypothesis 2 (H2).** *Instrumental images will be related to attractiveness toward an organization.*

### 3. Research Method

*3.1. Sampling*

This study used the critical informant method to collect survey data between January and September 2015 in South Korea. This study used two data sets collected from two different groups. In Study 1, participants were 1404 undergraduate students from three various universities in South Korea. Of the participants, 63.25% were male. The respondents aged between 20 and 25 comprised 69.73%.

In the pilot study phase, nine versions of a questionnaire were prepared, and participants received only one version. The nine companies targeted for the pretest questionnaire were Samsung Electronics, Korean Air, Asiana Airlines, KB Bank, KEPCO (Korea Electric Power Corporation), Hyundai Motor Company, POSCO, Samsung SDI, and SK Telecom. These companies represent some of the largest South Korean organizations, and many

undergraduates are interested in applying for jobs [37]. Participants were asked to indicate their perceptions of one company's symbolic and instrumental images and their attractiveness toward the organization as a workplace.

In Study 2, participants were 319 employees from nine different companies mentioned above. These companies were identified and targeted in the questionnaire of Study 1. The authors contacted employees of nine companies and requested voluntary participation in this survey. Across the nine companies, 21 to 48 employees from each company agreed to participate in the current study. Table 1 shows demographic information of the sample that participated in Study 2. Of all the participants, 68.65% were male (0.62% had unclear answers). Individuals in the age category between 31 and 39 years old comprised 49.22% of the participants.

**Table 1.** Demographic information (N = 319).

| Valuables | Index | Frequency | % |
|---|---|---|---|
| **Age** | 20 to 24 years old | 7 | 2.19 |
| | 25 to 30 years old | 71 | 22.26 |
| | 31 to 39 years old | 157 | 49.22 |
| | 40 to 49 years old | 78 | 24.45 |
| | 50 or older | 6 | 1.88 |
| **Education Level** | High School graduates | 26 | 8.15 |
| | Currently attending a 2-year college | 5 | 1.57 |
| | 2-year college graduates | 17 | 5.33 |
| | Currently attending a 4-year college | 14 | 4.39 |
| | 4-year college graduates | 194 | 60.50 |
| | Currently attending a graduate school | 5 | 1.57 |
| | Graduate degree holder | 59 | 18.50 |
| **Job Type** | Sales/Marketing | 55 | 17.24 |
| | Accounting/Finance | 11 | 3.45 |
| | HR/IR | 16 | 5.02 |
| | Institute of Technology | 42 | 13.17 |
| | Technical Support | 65 | 20.38 |
| | Customer Support | 62 | 19.44 |
| | Production | 5 | 1.57 |
| | Planning/Strategy | 40 | 12.54 |
| | Management Support/Secretary | 11 | 3.45 |
| | etc. | 12 | 3.76 |
| **Job Position** | Staff | 74 | 23.20 |
| | Chief | 18 | 5.64 |
| | An assistant manager | 108 | 33.86 |
| | A deputy general manager | 81 | 25.39 |
| | A department manager or higher | 38 | 11.81 |
| **Working years at the current organization** | Less than 1 year | 31 | 9.72 |
| | 1 to 3 years | 50 | 15.67 |
| | 4 to 6 years | 50 | 15.67 |
| | 7 to 9 years | 47 | 14.73 |
| | 10 or more years | 141 | 44.20 |

*3.2. Analytical Method and Measure*

3.2.1. Analytical Method

This study analyzed the data using a multi-step approach. In the first step, the measurement model was tested by subjecting the measures to a series of confirmatory factor analyses (CFAs). In the second step, a structural equation model was developed to evaluate the proposed hypotheses.

### 3.2.2. Measure

All items were measured on a 5-point rating scale, ranging from "strongly disagree" (1) to "strongly agree" (5).

#### Instrumental Images

This measurement model assessed the extent to which individuals perceived an organization to provide various functions. The study used a scale of eleven items proposed by Lievens and Highhouse [10] and Lievens and colleagues [22]. For example, some of the items were "ABC company offers the possibility to work together with different people," "ABC company offers job security," and "ABC company offers prospects for a certain future."

For Study 1, a CFA validated that a three-factor solution provided a good fit to the data ($\chi^2$ (41) = 451.17, normalized fit index (NFI) = 0.94, comparative fit index (CFI) = 0.95, incremental fit index (IFI) = 0.95, and standardized root mean square residual (SRMR) = 0.062). One-factor solution was not a good fit to the data ($\chi^2$ (44) = 036.25, NFI = 0.73, CFI = 0.74, IFI = 0.74, SRMR = 0.12). The three-factor solution fit the data much better than the one-factor solution ($\Delta\chi^2$ (3) = 1585.08, $p < 0.001$). We labeled the three factors as teamwork, advancement, and job security, in the given order.

For Study 2, the CFA ($\chi^2$ (41) = 253.48, NFI = 0.93, CFI = 0.94, IFI = 0.94, SRMR = 0.077) showed that a three-factor solution provided a good fit to the data. The one-factor solution was not a good fit to the data ($\chi^2$ (44) = 528.01, NFI = 0.73, CFI = 0.75, IFI = 0.75, SRMR = 0.12). The three-factor solution fit the data much better than the one-factor solution ($\Delta\chi^2$ (3) = 274.53, $p < 0.001$).

#### Symbolic Images

The study measured the following attributes of symbolic images of an organization: exemplary, sound, sincere, reliable, successful, confidential, pacesetting, competent, kind, affectionate, servable, unique, imaginative, adventurous, handsome, glamorous, and charming. The study established a seventeen-item scale from Aaker [31], Kim [38], Kim and Ahn [39], and Kim [40].

For Study 1, a CFA was included to assess the factor–item relationships. The five-factor solution provided a good fit to the data ($\chi^2$ (109) = 781.51, normed fit index (NFI) = 0.96, comparative fit index (CFI) = 0.97, incremental fit index (IFI) = 0.97, and standardized root mean square residual (SRMR) = 0.057. The one-factor solution or one-dimensional scale was not a good fit to the data ($\chi^2$ (119) = 8556.82, NFI = 0.66, CFI = 0.66, IFI = 0.66, SRMR = 0.15). The five-factor solution fit the data much better than the one-factor solution ($\Delta\chi^2$ (10) = 7775.31, $p < 0.001$). We labeled the five factors as sincerity, competence, affection, uniqueness, and sophistication.

For Study 2, the five-factor solution also fit the data much better than the one-factor solution ($\Delta\chi^2$ (10) = 1760.54, $p < 0.001$). The CFA showed that the five-factor solution provided a good fit to the data ($\chi^2$ (109) = 4.84, NFI = 0.95, CFI = 0.97, IFI = 0.97, SRMR = 0.070). The one-factor solution or one-dimensional scale was not a good fit to the data ($\chi^2$ (119) = 2165.38, NFI = 0.80, CFI = 0.81, IFI = 0.81, SRMR = 0.12).

#### Attractiveness Toward an Organization

This study adopted four items in the present study to measure large organizations' attractiveness in South Korea. The study used a scale proposed by Lievens and Highhouse [10], Highhouse, Lievens, and Sinar [41], Lievens, Van Hoye, and Schreurs [22]. Some of the items were "For me, ABC company would be a good place to work" and "ABC Company is attractive to me as a place of employment." For Study 1, the CFA was included to validate the scale ($\chi^2$ (2) = 120.68, NFI = 0.97, CFI = 0.97, IFI = 0.97, SRMR = 0.033), which shows that the scale is one-dimensional. For Study 2, the CFA validated the scale ($\chi^2$ (2) = 31.69, NFI = 0.97, CFI = 0.98, IFI = 0.98, SRMR = 0.031), showing that it is also one-dimensional. Table 2 shows the results of the confirmatory factor analysis (CFA).

**Table 2.** Confirmatory factor analysis (CFA).

|  | Variables | χ2 | NFI | CFI | IFI | SRMR |
|---|---|---|---|---|---|---|
| **Study 1** | Instrumental images (three factors) | 451.17 | 0.94 | 0.95 | 0.95 | 0.062 |
|  | Symbolic images (five factors) | 781.51 | 0.96 | 0.97 | 0.97 | 0.057 |
|  | Organizational attractiveness | 120.68 | 0.97 | 0.97 | 0.97 | 0.033 |
| **Study 2** | Instrumental images (three factors) | 253.48 | 0.93 | 0.94 | 0.94 | 0.077 |
|  | Symbolic images (five factors) | 4.84 | 0.95 | 0.97 | 0.97 | 0.070 |
|  | Organizational attractiveness | 31.69 | 0.97 | 0.98 | 0.98 | 0.031 |

(Note) Study 1: data from undergraduate students, Study 2: data from employees, NFI= Normalized Fit Index, CFI = Comparative Fit Index, IFI = Incremental Fit Index, SRMR = Standardized Root Mean Square Residual.

## 4. Analyses Results

### 4.1. Study 1 Findings

#### 4.1.1. Preliminary Findings

Intra-class correlation (ICC) was calculated to examine the extent to which the variance in attractiveness toward an organization was partitioned into the organization- and individual-level variances. ICC was 0.0039, indicating that 0.39% of the organization's variance in attractiveness was between organizations and 99.61% between individuals. The organization-level variance was 0.0035 and not significant ($\chi2$ (8) = 12.74, $p$ = 0.12). The ICC result indicated that there was no substantial organization-level variance in attractiveness toward an organization. In other words, attractiveness toward an organization was not affected by whether individuals indicated their organizational attractiveness about Samsung Electronics, Korean Air, or any other company. Thus, this study conducted only the one-level analysis without using multilevel modeling.

#### 4.1.2. Main Findings

A path analysis was conducted to test the model, as depicted in Figure 1. The fit was not good ($\chi2$ (8) = 481.74, NFI = 0.90, CFI = 0.90, IFI = 0.90, SRMR = 0.076). After removing non-significant paths and adding a non-hypothesized path between the first component of the instrumental image (i.e., the instrumental image of teamwork) and the second component of the instrumental image (i.e., the instrumental image of advancement) as suggested by the modification index, the revised model was examined again. The revised model provided a good fit to the data ($\chi2$ (13) = 184.45, NFI = 0.96, CFI = 0.96, IFI = 0.96, SRMR = 0.056), as shown in Figure 2.

As shown in Figure 2, some symbolic images, but not all, were related to the instrumental images, which then led to attractiveness toward an organization. For instance, the first component of the symbolic image (i.e., the symbolic image of being sincere) was related to the instrumental image of teamwork and job security but not related to the instrumental image of advancement. The symbolic image of being competent was positively related to the instrumental images of teamwork and job security. Surprisingly, however, the symbolic image of being competent was negatively related to the job security's instrumental image. The more competent the undergraduates consider an organization to be, the less job security the undergraduates perceive the organization to provide.

The symbolic image of being affectionate was only related to the instrumental image of job security. The more affectionate the undergraduates consider an organization to be, the more job security the undergraduates perceive the organization to provide. Interestingly, the instrumental image of teamwork was positively related to the instrumental image of advancement. The results showed that the more teamwork the undergraduates consider an

organization to provide, the more advanced the undergraduates perceived the organization to be.

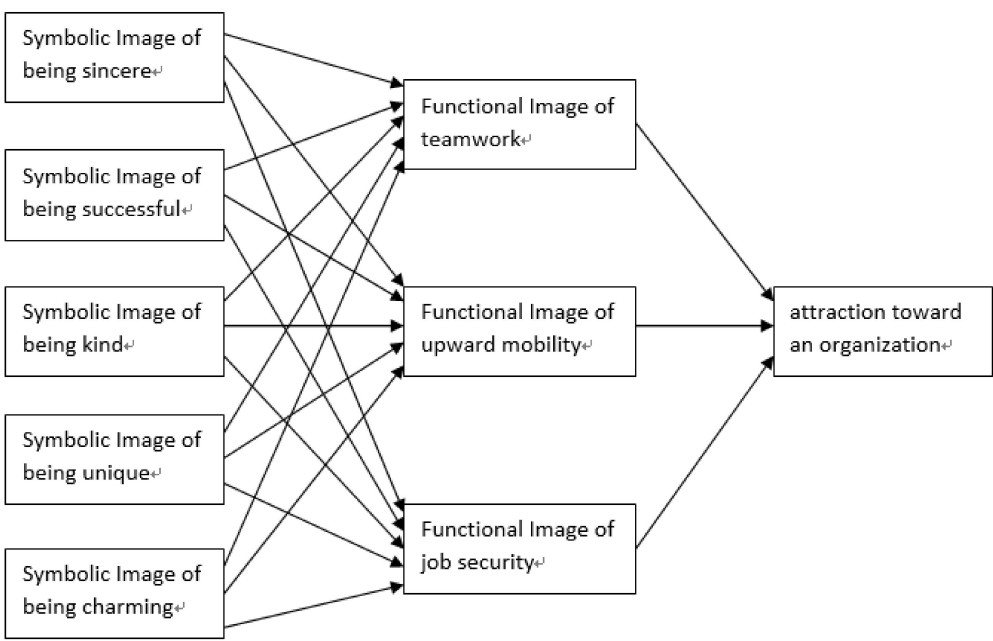

**Figure 1.** Original model. All arrows indicate hypothesized positive relationships.

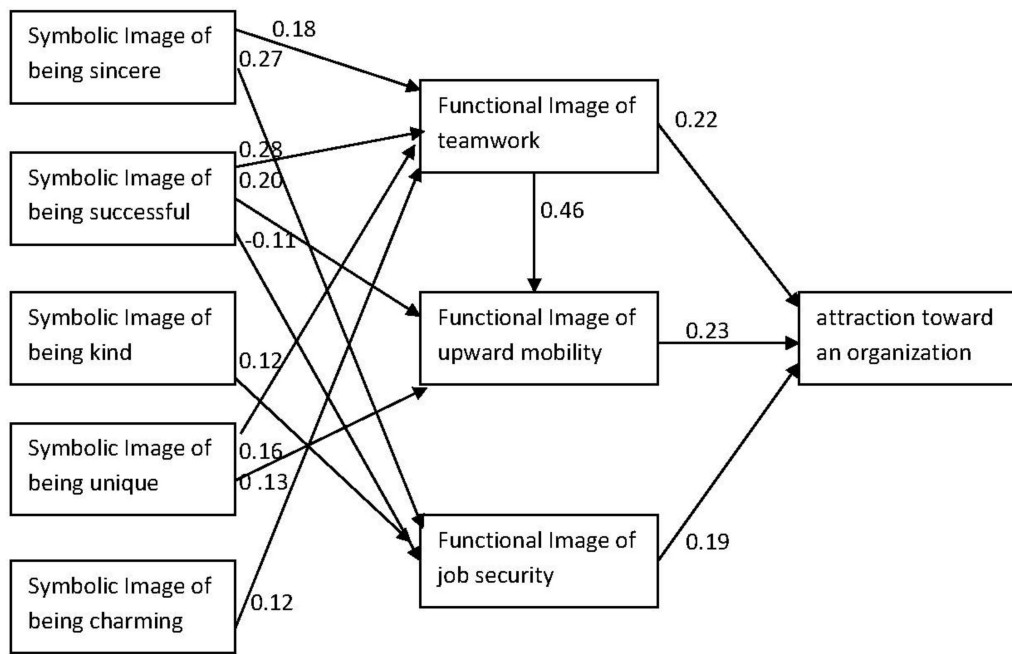

**Figure 2.** The revised model of Study 1 with potential applicants. All of the paths are significant at $p < 0.05$.

### 4.2. Study 2 Findings

#### 4.2.1. Preliminary Findings

Since employees of various companies might have indicated different levels of attractiveness toward their own companies, the ICC was calculated to examine the extent to which the organization level versus the individual level could explain the variance in attraction toward an organization. The ICC was 0.06, indicating that 6% of the variance in

attractiveness toward an organization was due to between-organization differences, and 94% was between-individual differences. Although the organization-level variance was 0.043, statistically significant ($\chi 2$ (8) = 27.18, $p$ = 0.001), the organization-level variance was not substantial. Because of the small variance attributable to organization-level differences, and the much larger and substantial variance attributable to individual-level differences, this study conducted the individual-level analysis without multilevel modeling.

### 4.2.2. Main Findings

A path analysis was conducted to test the model depicted in Figure 1. The fit was not satisfactory ($\chi 2$ (8) = 177.34, NFI = 0.93, CFI = 0.93, IFI = 0.93, SRMR = 0.085). The revised model tested in Study 1 was examined again. The revised model provided a better fit to the data ($\chi 2$ (13) = 79.18, NFI = 0.97, CFI = 0.97, IFI = 0.97, SRMR = 0.067), as shown in Figure 3.

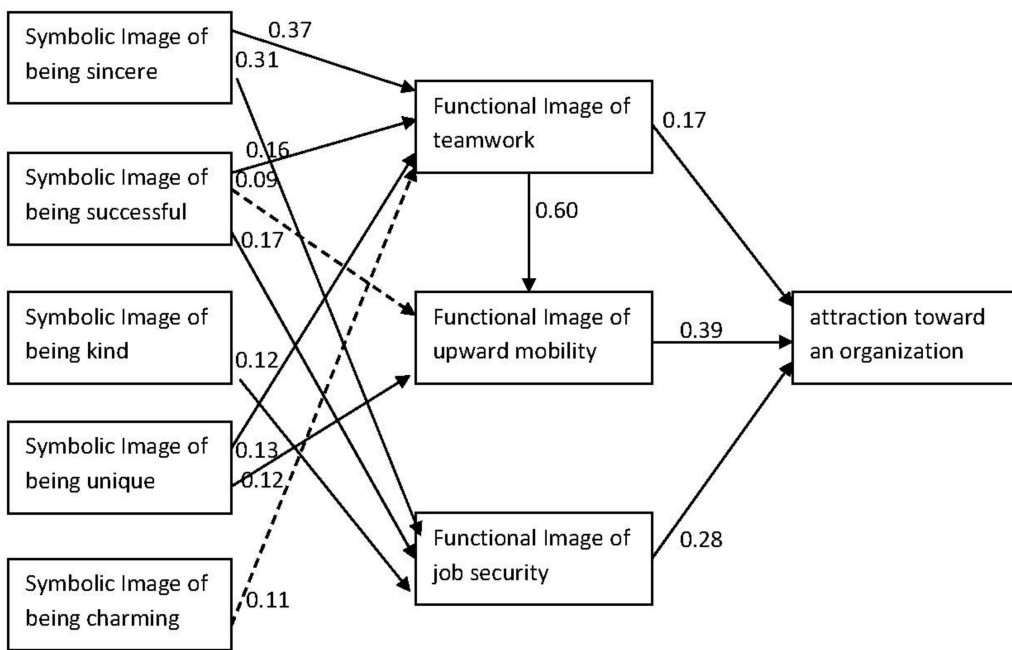

**Figure 3.** The final model of Study 2 with employee participants. The paths indicated with solid lines are significant at $p < 0.05$, and the paths with dotted lines are not significant.

As shown in Figure 3, most of the paths between the symbolic and instrumental images of Figure 2 were significant in Figure 3. However, the two paths were not significant. The symbolic image of being competent was not significantly related to the instrumental image of advancement, and the symbolic image of being sophisticated was not significantly related to the instrumental image of teamwork. These two paths were statistically significant in the revised model of Study 1 (Figure 2), but they were not significant in the model shown in Figure 3. This is probably due to the smaller sample size in Study 2 as compared to Study 1.

One notable difference between Figures 2 and 3 was that the relationship between the symbolic image of being competent and the instrumental image of job security was negative in Study 1 but positive in Study 2. This suggests that the more competent employees consider their organizations to be, the more job security they perceive their organizations to provide.

## 5. Discussion and Implications

### 5.1. Discussion

This study advances the understanding of the symbolic image of being affectionate of companies, using South Korean data. Previous studies [10,22,23,26] examined the image factors relation to applicants' perceived attractiveness and concentrated on individualistic

cultures. This study focused on collectivistic cultures. Several interesting conclusions can be drawn.

As a result, the symbolic image of being affectionate, which could be one of the primary characteristics in Korean culture, was discovered and related only to the job security's instrumental image. The more affectionate the potential applicants and employees consider an organization to be, the more job security they perceived the organization to provide. This finding signifies that Koreans perceive "Jeong (affection) culture" to reduce the possibility of downsizing and layoff.

Secondly, the instrumental image of teamwork is positively related to the instrumental image of advancement. The results show that the more teamwork the undergraduates and employees consider an organization to provide, the more advanced they perceive the organization to be. It is essential to achieve a team's goal, and employees could receive the incentive and earn a promotion, according to team performance in Korean organizations. This result implies that Korean employees could advance along with their team.

Thirdly, the instrumental image of job security is vital for both potential applicants and employees. According to data from Statistics Korea [42], the number of temporary workers reached 4.5 million in February 2018. When compared with February 2017, there is a 3.8% decrease. However, this number still goes up to almost 6 million when combined with daily workers. Moreover, there are more than 80% of temporary workers in some Korean companies such as Daewoo E&C (91.8%), LOTTE Engineering & Construction (81.5%), Hyundai E&C (81.4%), and Samsung Well-story (80.3%) (Kim, 2017). This result suggests that job security preference is due to the high percentage of temporary workers in Korea.

Fourthly, the symbolic image of being sincere is positively related to the instrumental image of teamwork and job security. The more sincere potential applicants and employees consider an organization to be, the more teamwork and job security they perceive the organization to provide.

Fifthly, the symbolic image of being competent is negatively related to the instrumental image of job security. The more competent applicants consider their organization to be, the less job security they perceive their organizations to provide. Most potential applicants seeking jobs in manufacturing companies prefer Samsung Electronics as their first company in Korea [43]. This study implies that the applicants recognize that Samsung Electronics' average length of service is below ten years. Samsung Electronics has many HR policies, such as non-smoking policies, performance-based systems, working night shifts, and providing a nonunion workplace. In other words, the potential applicants consider Samsung Electronics as the most competent company, but, at the same time, they also think that Samsung Electronics would not provide job security to them. This negative relationship may be why in 2016—unlike in 2013—potential applicants preferred Naver, CJ (Cheil Jedang), and Amorepacific more than Samsung Electronics [43]. Furthermore, supporting our supposition, applicants' reasons for choosing these companies included the likelihood and the vision of growth and development and the leading company image in the community and the business world. In contrast, choosing Samsung Electronics was most commonly due to the pride of being a member of the company.

Sixthly, employees in Korea perceive that the symbolic image of being competent is not related to the instrumental image of advancement. This result implies that the practice of getting promotions to seniority continues in Korean organizations. Moreover, they perceive that the symbolic image of being unique is related to the instrumental image of advancement. Many venture companies have non-seniority-based systems, but there are unique HR systems in Korea. For example, in Jennifer Soft Inc., an application performance management (APM) company, CEOs get together with employees for social parties and work side-by-side with them. Jennifer Soft Inc. (Paju, South Korea) was ranked first among other APM companies in Korea [44].

Their working hours include partaking in activities such as swimming and having coffee together [45]. The employees perceive that the organization provides opportunities for advancement because of this unique organizational image.

Finally, the instrumental image of teamwork, advancement, and job security was related to organizational attractiveness. This result supports previous studies' findings [7,10,22,29].

*5.2. Implications*

This study's results complement prior research claims [2,10,22,23,27,32,46] that have examined whether instrumental images are significant predictors for corporate attractiveness. However, this study's results are different from some of the previous claims. Symbolic images influence instrumental images.

Our research shows the importance and relationships of the symbolic and instrumental images for attractiveness toward an organization.

One of the results' implications is that corporate image perceptions are different for the potential applicants and employees. One of the notable differences between Figure 2 (potential applicants) and Figure 3 (employees) is that the relationship between the symbolic image of being competent and the instrumental image of job security is negative in Study 1 but positive in Study 2. The more competent employees consider their organization to be, the more job security they perceive their organization to provide. Moreover, the symbolic image of being competent is negatively related to the instrumental image of job security. This result partly supports the previous study [29].

Like Lievens' study [29], we also found the variance of groups of differences (potential applicants vs. employees).

However, unlike the previous study [29], we also found significant differences among the effect of symbolic images. These results provide some implications for the managers that corporate image perceptions are different for the potential applicants and employees. Therefore, a recruitment manager should consider this implication while developing a recruitment strategy.

Secondly, the symbolic image of being competent is positively related to the instrumental image of advancement in Study 1 but is not in Study 2. Many potential applicants perceive that the more competent they consider their organizations to be, the more advancement they perceive their organization to provide. Generally, employees perceive that there is no relationship between competence and advancement. These results mean that the actual employees do not perceive that competent organizations provide opportunities for advancement. This result may be caused by Korean culture that emphasizes authoritarianism and collectivism [47]. Such cultural factors and inconsistency of job order led to depressive symptoms in Korean workers [47]. Therefore, workers working in such unequal environments may feel less eager to believe in the idea that competence will lead to advancement, unlike applicants who may not know the organization's culture and employees yet. These results imply that the managers should consider providing opportunities for advancement while planning the motivation policies.

Thirdly, the symbolic image of being sophisticated is positively related to the instrumental image of teamwork in Study 1 but is not in Study 2. These results imply that employees do not perceive that sophisticated organizations provide opportunities for teamwork. Although it is hard to make a statement on its cause, it might be caused by unequal workplace culture or the environment employees have experienced.

Finally, our work has confirmed that this instrumental–symbolic framework can be applied to understand job seekers' attraction to organizations in a non-Western collectivistic culture. Therefore, more studies related to the instrumental–symbolic framework are needed to examine the applicability to other countries and cultures.

The different perceptions of corporate images between the potential applicants and employees give HR managers the strategy guidelines for recruitment and talent retention. Moreover, the instrumental images' results that are related to corporate attractiveness provide recruitment campaign guidelines for recruiters. More specifically, although most small- and medium-sized organizations could not provide high pay, they could give their employees more job security and advancement opportunities compared to large organizations.

These opportunities provided by the small- and medium-sized organizations would be attractive and meaningful for many applicants.

## 6. Limitations and Future Directions

Despite the new findings, a few limitations should be noted. First, it seems worthwhile to study integrated frameworks, including the various image factors. For instance, benefit systems, such as wages, locations, and support systems, and organizational characteristics, such as large-sized, decentralized, and multinational organizations [9], can be considered in future research models.

Secondly, since our study is based on one country, future studies could be conducted in different countries. Understanding the cross-cultural differences could be crucial for human resource managers to recruit global talents and attract their attention. Thirdly, since this study used cross-sectional data, future studies need to use longitudinal data to establish our model's causal claim empirically. Furthermore, this study focused on the symbolic and instrumental framework in Korea. However, the result showed significant differences between the perception of students and employees on specific images. Future studies should study the underlying reasons behind these differences.

Finally, this study focused on symbolic images, instrumental images, and attractiveness. However, Lievens et al. [9], for example, shows that some personalities moderate the relationship between organizational characteristics such as organizational size, internationalization, and organizational attractiveness. In other studies, openness to experience moderates the relationship between excitement and organizational attractiveness [32], and a social media presence improves their employer's brand image and attractiveness [48]. Therefore, future studies need to consider some moderating factors in the relationships between the image factors and organizational attractiveness.

**Author Contributions:** J.O. designed the research framework, conducted the survey, and analyzed the data; S.M. contributed to conceptualization, methodology, and editing. All authors have read and agreed to the published version of the manuscript.

**Funding:** This work was supported by the Ministry of Education of the Republic of Korea and the National Research Foundation of Korea (NRF-2019S1A5C2A0381234). This research was also funded by the Ministry of Trade, Industry, and Energy of the Republic of Korea (MOTIE—P0008703).

**Institutional Review Board Statement:** Ethical review and approval were waived for this study as the government funded project in Korea.

**Informed Consent Statement:** Informed consent was obtained from all subjects involved in the study.

**Conflicts of Interest:** The authors declare no conflict of interest.

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
