# Peer review of "What Matters for Job Security? Exploring the Relationships among Symbolic, Instrumental Images, and Attractiveness for Corporations in South Korea"

_sustainability, doi:10.3390/su13094854_

Round 1

Reviewer 1 Report

Really enjoyed your study. The perspective of a collectivistic culture is welcome, as we know since Hofstede's work that the dominant values of each culture can completely change organizational outputs and therefore,  human resources policies and practices.

I would like to see some latest references because in your bibliography 95% of the citations are previous to the last 4 yeas. I feel that would improve your work if you use some recent studies.

The results of empirical research could be a little more clear, maybe using some tables to sum up the results.

Author Response

Firstly, we thank the Respectful Referee for reviewing the manuscript, appreciation of its results, and valuable corrections and suggestions, which have been fully incorporated into the final manuscript's text.

  • The first Reviewer

1

Reviewer Comments

Really enjoyed your study. The perspective of a collectivistic culture is welcome, as we know since Hofstede's work that the dominant values of each culture can completely change organizational outputs and therefore, human resources policies and practices.

Author Response

Thank you very much for appreciation of the topic and the results of the paper

2

Reviewer Comments

I would like to see some latest references because in your bibliography 95% of the citations are previous to the last 4 years. I feel that would improve your work if you use some recent studies.

Author Response

Thank you for the useful comment and suggestion. We agree with the Respectful Referee. Therefore, we have added the following sentences.

=====================================================

Recently, Carpentier et al. (2019) found that the perceived communication characteristics of social media of companies for the potential applicants who are looking for an actual job posting are positively related to the organizational attractiveness and word-of-mouth based on the theory of symbolic attraction. Specifically, potential applicants who visited the company’s social media page can perceive the organizational warmth and competence and then feel the organizational attractiveness.

Kumari, S., & Saini, G. K. (2018) examined the effect of two instrumental images (career growth opportunities, work-life balance) and one symbolic image (CSR reputation) on the employer attractiveness. As a result, they found that all factors are related to on the employer attractiveness for potential candidates.

Waples, C. J., & Brachle, B. J. (2020) predicted that information of the organization's CSR activity would increase the organization's attractiveness and this hypothesis was supported by this research’s data. Specifically, job seekers prefer organizations which are doing CSR activities and included this information in the recruiting materials.

Carpentier, M., Van Hoye, & Weijters, B. (2019), Attracting applicants through the organization's social media page: Signaling employer brand personality, Journal of Vocational Behavior, Volume 115, pp. 1-14.

Kumari, S., & Saini, G. K. (2018), Do instrumental and symbolic factors interact in influencing employer attractiveness and job pursuit intention?, Career development international. , 23(4), p.444-462

Waples, C. J., & Brachle, B. J. (2020), Recruiting millennials: Exploring the impact of CSR involvement and pay signaling on organizational attractiveness. Corp Soc Resp Env Ma. 27:870–880

3

Reviewer Comments

The results of empirical research could be a little more clear, maybe using some tables to sum up the results.

Author Response

Thank you for the useful comment and suggestion. We agree with the Respectful Referee. We have added table 2(Confirmatory Factor Analysis (CFA)

Reviewer 2 Report

I want to thank the authors the opportunity to read your research.

The topic is interesting, mainly the relation between instrumental and symbolic attributes of the image of organizations. I'm not sure if the research fits the editorial scope of the journal, but I leave that questions for the editors.

The paper is well written and structured. It presents all the main informations to understand the research and even to replicate it. Nevertheless, I would like to share with you some thoughts in the form of commentaries and suggestions. My biggest concern is the timeliness of the research. Building a research based on references with more than 5 years (and no references form the last 2 years-3 years), and data collected more than 5 years ago, is not a good sign of being up-to-date.

  1. Line 48: please develop the idea. One sentence does not make a paragraph.
  2. Line 56 onwards: please develop a more organized and grounded justification for the differences between individualistic and collectivistic cultures. Do not use references from consumer behavior. Instead use support from the topic of organizational image
  3. Why the differences between potential applicants and employees? please support this option
  4. It's not clear what is the theoretical framework underlying your research; please make sure you make it clear
  5. The data is from 2015? If so, justify how and why it is still relevant 6 years later
  6. Of 44 references, less than 10 are from the last 5 years. And there is no references from 2019 onwards. Please make a serious update on references' list.

Author Response

Firstly, we thank the Respectful Referee for reviewing the manuscript, appreciation of its results, and valuable corrections and suggestions, which have been fully incorporated into the final manuscript's text

  • The Second Reviewer

1

Reviewer Comments

Building a research based on references with more than 5 years (and no references form the last 2 years-3 years), and data collected more than 5 years ago, is not a good sign of being up-to-date.

Of 44 references, less than 10 are from the last 5 years. And there is no references from 2019 onwards. Please make a serious update on references' list.

Author Response

Thank you for the useful comment and suggestion. We agree with the Respectful Referee. Therefore, we have added the following sentences.

=================================================

Recently, Carpentier et al. (2019) found that the perceived communication characteristics of social media of companies for the potential applicants who are looking for an actual job posting are positively related to the organizational attractiveness and word-of-mouth based on the theory of symbolic attraction. Specifically, potential applicants who visited the company’s social media page can perceive the organizational warmth and competence and then feel the organizational attractiveness.

Kumari, S., & Saini, G. K. (2018) examined the effect of two instrumental images (career growth opportunities, work-life balance) and one symbolic image (CSR reputation) on the employer attractiveness. As a result, they found that all factors are related to on the employer attractiveness for potential candidates.

Waples, C. J., & Brachle, B. J. (2020) predicted that information of the organization's CSR activity would increase the organization's attractiveness and this hypothesis was supported by this research’s data. Specifically, job seekers prefer organizations which are doing CSR activities and included this information in the recruiting materials.

2

Reviewer Comments

Line 48: please develop the idea. One sentence does not make a paragraph

Author Response

Thank you for the useful comment and suggestion. We agree with the Respectful Referee. Therefore, we have added the following contents.

. Specifically, if customers are feeling attractive to organizational images, they want to buy these corporates’ products. For example, Mr. Park, who runs a chicken restaurant in Seoul, became known when the franchise CEO Mr. Kim Kim released a letter on his SNS. A handwritten letter written by one high school student contains a message expressing gratitude to Mr. Park for serving chicken to him and his younger brother, who only cost 5,000 won last year. Afterwards, high school student's younger brother visited Park's chicken restaurant several times, and each time Park gave him chicken for free. When Mr. Park's story became known, consumers who wanted to give praise to Mr. Park ordered a lot because of Mr. Park’s good deeds [45].

3

Reviewer Comments

Line 56 onwards: please develop a more organized and grounded justification for the differences between individualistic and collectivistic cultures. Do not use references from consumer behavior. Instead use support from the topic of organizational image

Author Response

Thank you for the useful comment and suggestion. We agree with the Respectful Referee. Therefore, we have added the following sentences

=====================================================

Van Hoye and colleagues [2] examined that both instrumental (working conditions) and symbolic image (competence) were related to organizational attractiveness in a non-Western collectivistic culture and supported that the instrumental-symbolic framework should be applied differently across different countries, and cultures

4

Reviewer Comments

Why the differences between potential applicants and employees? please support this option

Author Response

Thank you for the useful comment and suggestion. We agree with the Respectful Referee. Therefore, we have added the following sentences.

====================================================

Managers should understand the changing demographics (i.e. baby boomers, generation X and generation Y or millennials) and the expectations of different generations (Lyons and Kuron, 2014). Previous studies show that young generation such as millennials like quick promotions (Ng et al., 2010; Smola and Sutton, 2002), flexibility in work hours, quality of life, recognition, and feedback (Cavazotte et al., 2012).

Lyons, S. and Kuron, L. (2014), “Generational differences in the workplace: a review of the evidence and directions for future research”, Journal of Organizational Behavior, Vol. 35, pp. S139-S157.

Ng, E., Schweitzer, L. and Lyons, S. (2010), “New generation, great expectations: a field study of the millennia generation”, Journal of Business Psychology, Vol. 25 No. 2, pp. 281-292.

Smola, K.W. and Sutton, C.D. (2002), “Generational differences: revisiting generational work values for the new millennium”, Journal of Organisational Behavior, Vol. 23 No. 4, pp. 363-382.

Cavazotte, F., Lemos, H.C. and Viana, M.D. (2012), “New generations in the job market: Renewed expectations or old ideals?”, Cadernos EBAPE. BR, Vol. 10 No. 1, pp. 162-180.

5

Reviewer Comments

It's not clear what is the theoretical framework underlying your research; please make sure you make it clear

Author Response

Thank you for the useful comment and suggestion. We agree with the Respectful Referee.

======================================================

Kumari, S., & Saini, G. K. (2018) examined the effect of two instrumental images (career growth opportunities, work-life balance) and one symbolic image (CSR reputation) on the employer attractiveness. As a result, they found that all factors are related to on the employer attractiveness for potential candidates.

Waples, C. J., & Brachle, B. J. (2020) predicted that information of the organization's CSR activity would increase the organization's attractiveness and this hypothesis was supported by this research’s data. Specifically, job seekers prefer organizations which are doing CSR activities and included this information in the recruiting materials. This study will use the instrument-symbolic framework which are related to organizational attractiveness as mentioned above.

6

Reviewer Comments

The data is from 2015? If so, justify how and why it is still relevant 6 years later

Author Response

Thank you for the useful comment and suggestion. We agree with the Respectful Referee.

======================================================

The issue of organizational image was a long-time subject in the field of HR studies. We developed the case of Korea based upon the previous theories and results. It took time to find out more insights and different perspectives to deal with this issue. We believe that this issue has been a steady topic in HR studies, and therefore the cross-section survey study is still worthwhile for about five years.

We thank the Respectful Editors and Respectful Referee of the paper very much for the kind care about the article and its results,

It is an honor for us to have our work published in the sustainability

With kind regards,

Round 2

Reviewer 2 Report

Thank for your effort in improving the paper. Although you managed to improve the paper following the suggestions, I am still concerned with the a research based on data from 6 years ago. Stating that "this issue has been a steady topic in HR studies" is hardly an argument for disclose a research work with 6 years. But I will leave that decision to the Editor.